# Magnitude and factors associated with post-tuberculosis lung disease in low- and middle-income countries: A systematic review and meta-analysis

**Elizabeth Maleche-Obimbo**[1]\*, **Mercy Atieno Odhiambo**[2], **Lynette Njeri**[3], **Moses Mburu**[4], **Walter Jaoko**[5], **Fredrick Were**[1], **Stephen M. Graham**[6]

**1** Department of Paediatrics & Child Health, University of Nairobi, Nairobi, Kenya, **2** Neurosciences Unit, KEMRI-Wellcome Trust Research Programme, Kilifi, Kenya, **3** School of Medicine, University of Nairobi, Nairobi, Kenya, **4** Clinical Trials Facility, KEMRI-Wellcome Trust Research Programme, Kilifi, Kenya, **5** Department of Medical Microbiology & Immunology, University of Nairobi, Nairobi, Kenya, **6** Department of Paediatrics, University of Melbourne, Melbourne, Australia

\* elizabeth.obimbo@uonbi.ac.ke

## Abstract

### Background

Emerging evidence suggests that after completion of treatment for tuberculosis (TB) a significant proportion of patients experience sequelae. However, there is limited synthesized evidence on this from low-income countries, from Sub-Saharan Africa, and in HIV infected individuals. We seek to provide an updated comprehensive systematic review and meta-analysis on the magnitude and factors associated with post-TB lung disease (PTLD) in low- and middle-income countries (LMICs).

### Methods

We searched PubMed, Embase and CINAHL for studies from LMICs with data on post-TB lung health in patients who had previously completed treatment for pulmonary TB. Data on study characteristics, prevalence of PTLD–specifically abnormal lung function (spirometry), persisting respiratory symptoms and radiologic abnormalities were abstracted. Statistical analysis was performed using Microsoft Excel and R version 4.1 software, and random effects meta-analysis conducted to compute pooled prevalence of PTLD, evaluate heterogeneity, and assess factors associated with PTLD.

### Results

We identified 32 eligible studies with 6225 participants. Twenty-one studies were from Africa, 16 included HIV infected participants, spirometry was conducted in 20 studies, symptom assessment in 16 and chest imaging in eight. Pooled prevalence of abnormal lung function was 46.7%, persistent respiratory symptoms 41.0%, and radiologic abnormalities 64.6%. Magnitude of any type of PTLD varied by HIV status (HIV- 66.9%, HIV+ 32.8%, p = 0.0013), across geographic setting (SE Asia 57.5%, Southern America 50.8%, and

**Data Availability Statement:** All data relating to this article has been submitted as part of the Supporting information.

**Funding:** The authors received no specific funding for this work.

**Competing interests:** The authors have declared that no competing interests exist.

Africa 38.2%, p = 0.0118), and across urban-rural settings (symptom prevalence: rural 68.8%, urban 39.1%, mixed settings 27.9%, p = 0.0035), but not by income settings, sex or age-group.

## Conclusions

There is high burden of post-TB persistent respiratory symptoms, functional lung impairment and radiologic structural abnormalities in individuals living in LMICs. Burden varies across settings and by HIV status. This evidence may be valuable to advocate for and inform implementation of structured health care specific to the needs of this vulnerable population of individuals.

## Introduction

Tuberculosis (TB) is a common disease caused by *Mycobacterium tuberculosis (MTb)*, and infection is largely transmitted from person to person through inhalation of infected air droplets. The most common site of initial infection and disease is the lungs and spread to other intrathoracic tissues such as hilar lymph nodes and pleura occurs frequently. TB is among the top ten causes of morbidity and mortality globally, and over the past decade has been the leading cause of death from a single infectious agent. and in 2019 an estimated 1.2 million people died, and an estimated 10 million people fell ill with TB [1]. Effective treatment for TB is available through national TB programs in most of the world, and national programs report high treatment success rates for people treated with first-line regimens—86% overall and ranging from 74% to 91% across World Health Organisation (WHO) regions. Based on modelling of these global epidemiologic TB survival data, it is estimated that in 2020 there were approximately 155 million TB survivors globally, of whom 18% had TB within the preceding five years [2, 3].

There is an increasing body of evidence suggesting that after completion of TB treatment, despite clearance of TB bacilli, a significant proportion of TB survivors are left with post-TB sequelae, possibly due to damage of lung tissues during the period of active TB disease [4]. Chronic inflammation in response to persistent MTb infection, tissue damage and destructive healing responses are known pathophysiologic effects of TB disease, with resultant damaged airways, lung parenchyma or pleura that persists after completion of TB treatment [5, 6].

Most of the reviews within the past decade have focused on airway pathology and chronic obstructive pulmonary disease (COPD) following TB disease mainly in middle- and high-income countries, revealing a clear positive association between TB and COPD [7–9]. One narrative systematic review by Meghji et al reported varying prevalence of structural abnormalities seen in chest imaging after completion of TB treatment, and noted limited published data on post-TB sequelae in Sub-Saharan Africa, and among people living with Human Immunodeficiency Virus (HIV) [10].

These recent reviews on sequelae of TB have provided better insight into the magnitude of the problem, however they noted the paucity of evidence on post-TB lung disease (PTLD) from low-income countries, from Africa and in persons at high risk of severe TB such as children and people living with HIV [11]. There have been few reviews that examine the important patient centred outcome measure of persistent respiratory symptoms following treatment for TB.

Current TB treatment programs provide well-structured care up to completion of TB treatment, and discharge the patients from care, assuming treatment success, but no structured

care thereafter [12]. Over the past three years the global scientific community has convened to collaboratively examine the problem of sequelae from pulmonary TB and noted the paucity of quality research on individuals surviving TB. The expert panels developed consensus definitions for PTLD and specific clinical standards for the assessment and management of PTLD [13]. They highlighted the need for increased awareness about PTLD, and the urgent need for research to generate evidence to guide the development of solutions for former TB patients who have sequelae [13, 14].

In this systematic review and meta-analysis, we seek to provide an updated comprehensive synthesis of emerging evidence from low- and middle-income countries on the magnitude and factors associated with PTLD, we examine the patient-centred outcome of persistent respiratory symptoms, as well as functional lung impairment and structural abnormalities from imaging.

## Methods

This review adhered to the Preferred Reporting Items of Systematic review and Meta-analysis (PRISMA) guidelines and the protocol was registered on PROSPERO under the registration number: CRD42021241714(https://www.crd.york.ac.uk/prospero/).

### Literature search

PubMed, Embase and CINAHL were searched for studies including patients with previous pulmonary or intrathoracic TB to outline the magnitude of lung sequelae following completion of TB treatment, and associated risk factors. We searched for studies from inception to 11[th] March 2021 and updated the database search on the 17[th] of November 2021. The search terms "Estimate" AND "Magnitude" AND "Post Tuberculosis Lung disease" AND "LMICs" were combined during the database search. Alternative terms were inputted into the search strategy using these key words (S1 Table). Handsearching of included articles and relevant review papers was conducted to identify additional potentially eligible articles.

### Eligibility criteria

The population of interest was patients who had ever been diagnosed with pulmonary tuberculosis disease, or intrathoracic TB disease, and assessed for post-TB lung disease (PTLD) after completion of TB treatment. Patients must have been living in low- or middle-income countries as classified by the World Bank [15]. Magnitude of PTLD was defined broadly to include prevalence or incidence of lung sequelae following pulmonary TB disease. The outcomes of interest were the magnitude of PTLD whether expressed as prevalence or incidence, and factors associated with PTLD.

Articles were deemed eligible for inclusion if they were cross-sectional studies providing prevalence data or cohort studies providing incidence or period prevalence data on post-TB lung health, and any sample size of study was included provided key outcome data were reported. Acceptable publications included scientific journal articles and thesis manuscripts. We included conference proceedings or abstracts, scientific meeting proceedings and reports, and brief communications if they contained sufficient outcome data. Only articles written in English, or those with an English translation were included. Articles that did not report the outcomes of interest and those that had a highly biased selection criteria based on consensus by the review authors were excluded. Studies without English translation, literature reviews, systematic reviews, case reports and case series were also excluded.

**Case definitions.** The exposure prior TB was based on documented prior TB diagnosis and prior TB treatment either from patient medical record or patient report. Details of how

the TB diagnosis in each specific study was made were captured where reported and included any combination of clinician diagnosis based on symptoms and signs of TB, radiologic, bacteriologic diagnosis, or a combination of these. A minimum definition of PTLD included "evidence of chronic respiratory abnormality, with or without symptoms, attributable at least in part to previous pulmonary tuberculosis" [14] and included any other presentations of lung disease or impairment after completion of TB treatment which were deemed appropriate as assessed by the authors.

## Study selection, data extraction and study appraisal

A multi-stage screening method was adopted for study selection. First, all the titles and abstracts were screened against the eligibility criteria; followed by screening of the full texts of potentially relevant studies. Screening and selection of studies were conducted independently by two reviewers (M.A and L.N) and discrepancies were resolved through consensus or consultation with a third reviewer, an expert in this field (E.M.O.). A Prisma flow chart was developed to describe the screening and eligible study identification results.

Data relevant to the objectives was extracted into electronic piloted standardised forms that had been developed in Microsoft Excel software (Microsoft Corporation, 2019). Data collected included study characteristics such as author(s), year of publication, setting, country, continent, World Health Organisation (WHO) regional classification, study design, World Bank economic classification, sample size, and patient specific characteristics such as age, sex, and HIV status of participants. Outcome data abstracted included number of patients assessed for outcome of interest, number of patients with outcome (respiratory symptoms, abnormal spirometry or abnormal radiology), as well as incidence rates, prevalence rates, and risk ratios or odds ratios for factors associated with PTLD if reported. Data collection was conducted by two review authors (M.A. and L.N.) and disagreements were resolved through consensus or consultation with a third author (E.M.O.). The third reviewer and principal investigator (E.M.O.), a clinical expert in the field of respiratory medicine, also independently reviewed all the clinical data for the included studies.

The Newcastle-Ottawa Scale was used to assess methodological quality of included studies [16]. The focus was on selection—case and non-case definitions and representativeness (having the outcome of interest); comparability—criteria used to decide on the cases and comparison group and matching, if any; and ascertainment of exposure and non-response rate. Included studies were scored out of 10 as per the quality appraisal scale. Those that scored between 0 to 3 were considered poor quality, 4 to 6 were rated average quality and studies that scored 7 and above were rated good quality.

## Statistical analysis

Statistical analysis was performed using Microsoft Excel (Microsoft Corporation, 2019) and R 4.1.2 software. A quantitative synthesis and meta-analysis was conducted to determine pooled prevalence of abnormal lung function, of persistent respiratory clinical symptoms, and of abnormal radiologic findings post-TB using meta R package. Forest plots were generated to illustrate the findings, and corresponding funnel plots generated using dmeta package when ten or more studies were available for a specific PTLD outcome. Heterogeneity between studies was assessed using the $I^2$ statistic. An $I^2$ value of >50% was considered high and a random effects model meta-analysis was then applied in such cases to account for heterogeneity. Residual heterogeneity was assessed using sensitivity analyses. To evaluate factors associated with PTLD, we conducted subgroup meta-analyses stratified by key factors of interest–geographic region, World Bank economic classification, urban-rural settings, and patient specific factors:

HIV status, age-group, and sex where possible. Cochran's Q test was applied to test for significant difference between sub-groups [17, 18].

## Results

A total of 3934 studies were identified following database searches and 54 studies from hand-searching reference lists of included studies (Fig 1). After removal of duplicates, 3974 titles were reviewed for inclusion and 1715 found ineligible. We then reviewed 2259 abstracts and identified 102 potentially eligible studies which we subjected to full paper assessment for eligibility. Finally, 32 studies fulfilled all eligibility criteria and were included in the review [19–50]. The excluded studies (n = 70) at full paper assessment either did not provide sufficient outcome data on post-TB lung disease to enable determination of PTLD prevalence (n = 30), were from high-income countries (n = 18), reported on patients with active TB (n = 8) or the inclusion criteria selected a highly biased group of participants, specifically included only those with lung disease/sequelae (n = 4). In addition, we excluded two review papers and one editorial from which we could not extract any individual participant data. Finally, we were unable to access the full papers of two studies while another two studies did not have an English translation of the full paper available online.

### Description of studies

Regionally, majority of the included studies (n = 21) were from the African region, followed by seven studies from Southeast Asia [22, 24, 25, 27, 32, 39], two studies from the Americas (both from South America) [23, 31] and one study each from the Eastern Mediterranean [34] and

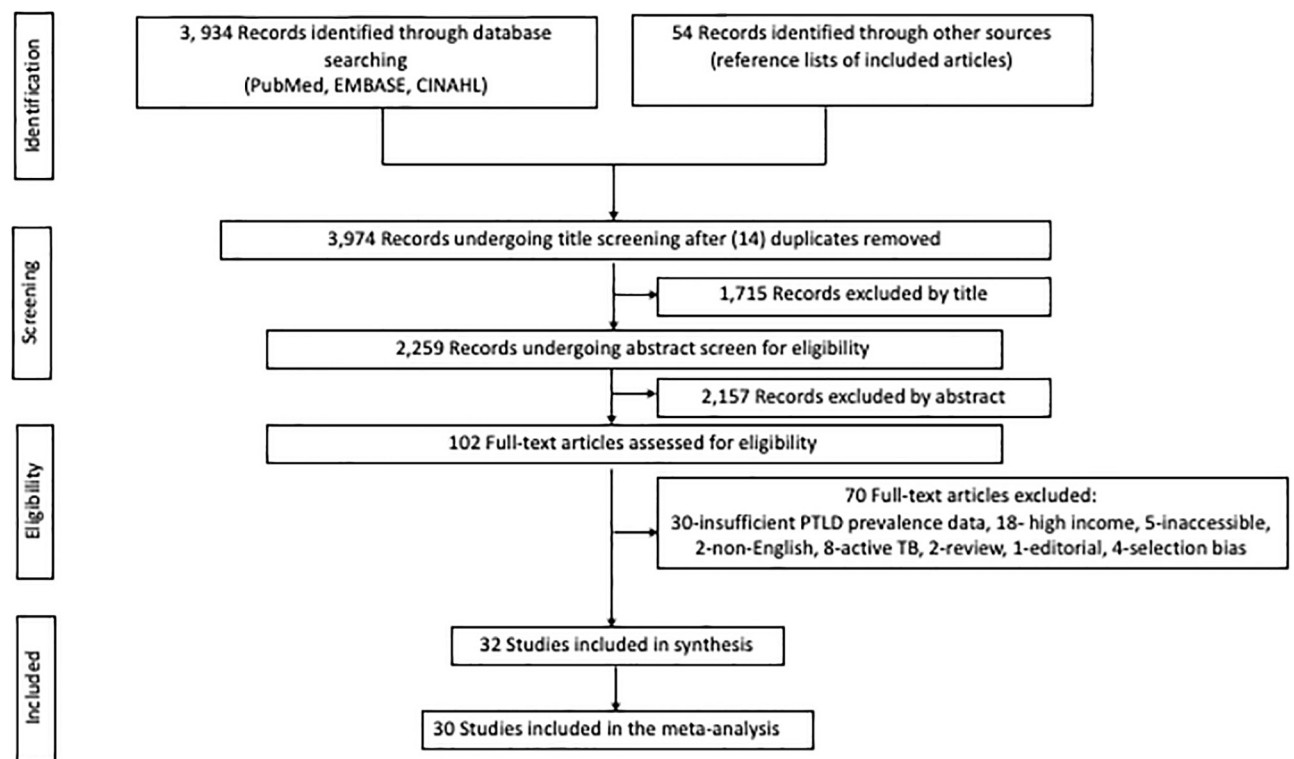

**Fig 1. PRISMA flow chart.** A flow chart showing study identification and selection process.

Western Pacific regions [26] (Table 1a). India and South Africa each contributed five studies, Cameroon, Uganda, and Tanzania each contributed two, while Tunisia, Zimbabwe, Kenya, Malawi, Mozambique, Nigeria, Philippines, Indonesia, Ethiopia, Democratic Republic of Congo, Benin, and Brazil each contributed one study. Nine studies were conducted in low-income countries, 17 in low-middle income countries, and six were conducted in upper-middle income countries.

The 32 studies had varying sample sizes ranging from 18 to 798 participants, and we computed a median sample size of 96 (interquartile range [IQR] 65–260) study participants, with a total of 6225 prior TB patients across the studies (Table 1a). Of note, in studies that had a combination of patients who had prior TB and other patients who did not have history of TB we abstracted data only on the subset of patients with our exposure of interest (prior TB) for this review.

Most studies (n = 19) provided information of the method used for prior TB diagnosis where; nine (28%) included only bacteriologically confirmed patients, six (19%) included patients diagnosed using a combination of clinical, bacteriologic, and radiological methods, three (9%) used a combination of clinical and bacteriologic methods and one (3%) study utilized radiologic evidence only (Table 1b).

Thirteen studies (41%) did not provide details on the method used for prior TB diagnosis among the patients. Among included studies, six included a mixture of first time TB and retreatment TB cases. Two studies exclusively enrolled patients with prior multi-drug resistant TB (MDR-TB) while most studies (n = 24) evaluated patients with prior first time TB, either clinically diagnosed or confirmed drug-susceptible TB.

The majority of studies (n = 20) used one method of assessment for sequelae. Specifically, 12 did spirometry only, six assessed respiratory symptoms only, and two did radiology only. Of the remaining 12 studies, four studies reported on all three outcome measures, seven studies reported on two outcome measures, and one [34] did not specify which measure was used.

Eighteen studies documented timing of PTLD evaluation providing either median/mean time to assessment or range (minimum—maximum time). Post-TB assessment was done within 6 months (n = 7), between 7–12 months (n = 4), between 24–36 months (n = 5), or 60 + months (n = 2) after completion of TB treatment (Table 1b).

The majority of the study populations were exclusively adults (n = 23 studies); one study was conducted exclusively in children [34] while eight studies included both adolescent and adult populations (Table 1b). Median or mean age was reported in 25 studies and ranged from 7.5–41.5 years (median) or 27.6–51.8 years (mean). Twenty-three studies reported on the sex of study participants with females comprising between 2% to 91% of participants. HIV status was reported in 19 studies as follows—six studies included only HIV infected study participants, three included only HIV uninfected participants, while 10 studies included mixed HIV infected and uninfected study participants with HIV positivity ranging from 3% to 91% (Table 1b).

Nearly half of the studies (n = 15) were of good quality (score 7 and above) based on the Newcastle Ottawa scale assessment, 16 were rated average quality and only one study scored less than 4 during methodological assessment of the included studies (Table 1b).

## Prevalence of post-TB lung disease

Prevalence of PTLD was reported in 6,225 patients from the 32 studies, of which two studies involved 80 MDR-TB survivors. After reviewing the data, we opted to report the two MDR-TB studies separately in narrative format and exclude the 80 MDR-TB survivors from the meta-analyses, since drug resistant TB differs significantly as an exposure from drug-susceptible

**Table 1.** a: Characteristics of included studies—geographic distribution, income level, study design, setting and sample size. b: Characteristics of included studies–diagnostic criteria for prior TB and post-TB lung disease; age, sex, and HIV status of study participants.

**a**

| No. | First Author (Year) | Country | WHO Region | Income* | Setting | Study Design | Urbanization | Sample | |
|---|---|---|---|---|---|---|---|---|---|
| 1 | Plit M (1998) | South Africa | African | UMI | Hospital | Prospective cohort | Urban | 74 | |
| 2 | De Valliere (2004) | South Africa | African | UMI | Hospital | Prospective cohort | Rural | 34 | |
| 3 | Ehrlich (2004) | South Africa | African | UMI | Community | Cross-sectional | Mixed | 326 | |
| 4 | Muniyandi (2007) | India | South East Asian | LMI | Hospital | Retrospective cohort | Mixed | 436 | |
| 5 | Cruz Rde (2008) | Brazil | Americas | UMI | Hospital | Prospective Cohort | Urban | 96 | |
| 6 | Maguire (2009) | Indonesia | South East Asian | LMI | Hospital | Prospective cohort | Urban | 69 | |
| 7 | Banu Rekha (2009) | India | South East Asian | LMI | Hospital | Cross-sectional | Urban | 198 | |
| 8 | Idolor (2011) | Philippines | Western Pacific | LMI | Community | Cross-sectional | Rural | 26 | |
| 9 | Akkara (2013) | India | South East Asian | LMI | Hospital | Cross-sectional | Rural | 257 | |
| 10 | Pefura-Yone (2014) | Cameroon | African | LMI | Hospital | Cross-sectional | Urban | 177 | |
| 11 | Akanbi (2015) | Nigeria | African | LMI | Hospital | Cross-sectional | Urban | 82 | |
| 12 | Binegdie (2015) | Ethiopia | African | LI | Hospital | Cross-sectional | Urban | 725 | |
| 13 | De la Mora (2015) | Mexico | Americas | LMI | Hospital | Cross-sectional | Urban | 70 | |
| 14 | Menon (2015) | India | South East Asian | LMI | Hospital | Retrospective cohort | Urban | 441 | |
| 15 | Mbatchou Ngahane (2016) | Cameroon | African | LMI | Hospital | Cross-sectional | Urban | 269 | |
| 16 | Snène H (2016) | Tunisia | Eastern Mediterranean | LMI | Hospital | Retrospective Cohort | Urban | 33 | |
| 17 | Attia (2018) | Kenya | African | LMI | Hospital | Cross-sectional | Urban | 96 | |
| 18 | Katoto (2018) | DR Congo | African | LI | Community | Cross-sectional | Rural | 441 | |
| 19 | Magitta (2018) | Tanzania | African | LMI | Community | Cross-sectional | Rural | 18 | |
| 20 | North (2018) | Uganda | African | LI | Hospital | Cross-sectional | Rural | 18 | |
| 21 | Singla (2018) | India | South East Asian | LMI | Hospital | Cross-sectional | Urban | 46 | |
| 22 | Van Kampen (2019) | Uganda | African | LI | Community | Cross-sectional | Mixed | 798 | |
| 23 | Chin (2019) | Zimbabwe | African | LMI | Hospital | Prospective cohort | Urban | 175 | |
| 24 | Fiogbe (2019) | Benin | African | LMI | Hospital | Cross-sectional | Urban | 189 | |
| 25 | Gupte (2019) | India | South East Asian | LI | Hospital | Prospective cohort | Mixed | 172 | |
| 26 | Osman (2019) | South Africa | African | UMI | Hospital | Cross-sectional | Urban | 51 | |
| 27 | Kayongo (2020) | Uganda | African | LI | Hospital | Cross-sectional | Rural | 66 | |
| 28 | Meghji (2020) | Malawi | African | LI | Hospital | Prospective cohort | Urban | 405 | |
| 29 | Khosa (2020) | Mozambique | African | LI | Hospital | Prospective cohort | Urban | 62 | |
| 30 | Auld (2021) | South Africa | African | UMI | Hospital | Prospective cohort | Urban | 85 | |
| 31 | Ddungu (2021) | Uganda | African | LI | Hospital | Cross-sectional | Urban | 72 | |
| 32 | Mpagama (2021) | Tanzania | African | LMI | Hospital | Cross-sectional | Mixed | 219 | |

**b**

| Total No. | 32 | 32 | 32 | 32 | 32 | 32 | 32 | 6,225 | |
|---|---|---|---|---|---|---|---|---|---|
| **No.** | **First Author name (Year)** | **Prior TB Definition** | **PTLD Definition[a]** | **Age included** | **Age Group** | **Age in yr.** | **Sex** | **HIV Status** | **Quality Score** |
| | | | | | | Mean (SD) or Median (IQR) | (% female) | (% HIV +) | |
| 1 | Plit M (1998) | Clin, bact, radiol, | Spiro | >18 | Adults | 35 (10) | 30% | 18% | 6 |
| 2 | De Valliere (2004) | Bact | Spiro, radiol | >18 | Adults | 40.2 | 91% | 9% | 5 |

*(Continued)*

**Table 1.** (Continued)

| | | | | | | | | | |
|---|---|---|---|---|---|---|---|---|---|
| 3 | Ehrlich (2004) | NR | Clin | >15 | Adolescents, adults | NR | 2% | NR | 7 |
| 4 | Muniyandi (2007) | Clin, bact | Clin | NR | Adults | NR | 34% | NR | 3 |
| 5 | Cruz R (2008) | NR | Spiro, radiol | >15 | Adolescents, adults | 41.08 (14.32) | 46% | 0%[c] | 5 |
| 6 | Maguire (2009) | Bact | Spiro, clin | >18 | Adults | 30.6 (23.9–38.9) | 33% | 3% | 8 |
| 7 | Banu Rekha (2009) | Bact | Spiro, clin, radiol | >18 | Adults | 46 (27) | 37% | NR | 6 |
| 8 | Idolor (2011) | NR | Spiro | >40 | Adults | NR | NR | NR | 7 |
| 9 | Akkara (2013) | NR | Spiro, clin, radiol | >18 | Adults | NR | 26% | NR | 5 |
| 10 | Perfura-Yone (2014) | Bact | Spiro, clin | >18 | Adults | 32.0(24.0–45.5) | 43% | 27% | 4 |
| 11 | Akanbi (2015) | NR | Spiro, clin | >30 | Adults | 44.5 (7.1) | NR | 100% | 6 |
| 12 | Binegdie (2015) | Radiol | Radiol | >12 | Adults | 40 | 9% | NR | 7 |
| 13 | De la Mora (2015) | NR | Spiro, clin, radiol | >24 | Adults | NR | NR | NR | 6 |
| 14 | Menon (2015) | Clin, bact, radiol | Radiol | >18 | Adults | NR | 36% | NR | 5 |
| 15 | Mbatchou Ngahane (2016) | Clin, bact, radiol | Spiro | >15 | Adolescents, adults | 34.2 (9.07) | 46% | 18% | 7 |
| 16 | Snène H (2016) | Clin, bact, radiol | NR | <18 | Children | 7.5 | NR | NR | 5 |
| 17 | Attia (2018) | NR | Spiro, radiol | >10–19 | Adolescents, adults | 39.0 (32.0–45.0) | NR | 100% | 6 |
| 18 | Katoto (2018) | NR | Clin | >15 | Adolescents, adults | 44.6 (14.9) | 32% | NR | 7 |
| 19 | Magitta (2018) | NR | Spiro, clin | >35 | Adults | 51.8 (10.0) | NR | NR | 8 |
| 20 | North (2018) | NR | Spiro, clin | >40 | Adults | 52.0 (48.0–55.0) | NR | 100% | 6 |
| 21 | Singla (2018) | Bact | Spiro, clin, radiol | 1—>45 | Adolescents, adults | 27.6 (10.5) | 46% | NR | 6 |
| 22 | Van Kampen (2019) | NR | Clin, radiol | >15 | Adolescents, adults | NR | 47% | NR | 9 |
| 23 | Chin (2019) | Clin, bact | Spiro, clin, radiol | NR | Adults | 41.0 (33.0–48.0) | 42% | 65% | 6 |
| 24 | Fiogbe (2019) | Clin, bact, radiol | Spiro, clin | >18 | Adults | 37.0 (30.0–47.0) | 32% | 11% | 10 |
| 25 | Gupte (2019) | Clin, bact | Spiro | >18 | Adults | 32.0 (23.0–39.0) | 48% | 4% | 6 |
| 26 | Osman (2019) | Bact | Clin | >18 | Adults | 40.0 (31.0–49.0) | 37% | 51% | 9 |
| 27 | Kayongo (2020) | NR | Spiro, clin | >35 | Adults | 48 | NR | 100% | 8 |
| 28 | Meghji (2020) | Clin, bact, radiol | Spiro, clin, radiol | >15 | Adolescents, adults | 35.0 (28.0–41.0) | 32% | 60% | 8 |
| 29 | Khosa (2020) | Bact | Spiro | >18 | Adults | 29.5 (25.0–40.0) | 32% | 63% | 9 |
| 30 | Auld (2021) | Bact | Spiro, clin | >18 | Adults | 36.0 (31.0–43.0) | 67% | 100% | 4 |
| 31 | Ddungu (2021) | NR | Spiro, clin | >30 | Adults | 45 (1) | NR | 100% | 7 |
| 32 | Mpagama (2021) | Bact | Spiro, clin | >18 | Adults | 45 (35–55) | 12% | 16% | 7 |
| No. Studies | **32** | **20** | **31** | **30** | **32** | **25** | **23**[b] | **19**[c] | **32** |

*World Bank economic classification: LI = low-income; LMI = lower-middle income; UMI = upper-middle income

[a] Clin = clinical assessment, Bact = bacteriologic, Radiol = radiologic, Spiro = spirometry. NR = not reported.

[b] mean = 32%.

[c] mean = 42%

forms of TB. The following section focuses on data from 30 studies which did not involve post MDR-TB patients.

**Prevalence of abnormal lung function following TB.** Spirometry was conducted in 2701 study participants in 20 studies (excludes MDR TB studies), and 1347 (49.9%) patients had

impaired lung function. Prevalence of abnormal spirometry varied across the individual studies and ranged between 18.3 and 86.8% (Table 2). Meta-analysis revealed the pooled prevalence to be 46.7% (95% CI: 36.9–56.7%), with significant heterogeneity across studies ($I^2$ = 95%, t2 = 0.6843, p<0.01) (Fig 2).

**Prevalence of persistent respiratory symptoms following TB.** Persisting respiratory symptoms after completion of TB treatment was reported in 3811 patients from 16 studies, and 1533 (40.2%) remained symptomatic. Prevalence of persisting symptoms after completion of TB treatment varied across individual studies and ranged between 12.9 and 84.8% (Table 2). Meta-analysis revealed the pooled prevalence to be 41.0% (95% CI: 29.4–53.7%), with significant heterogeneity ($I^2$ = 97%, t2 = 0.8773, p<0.01) (Fig 3).

**Table 2.  Prevalence of post-TB lung disease–abnormal lung function, persistent symptoms, or abnormal radiology.**

| Study Index No. | Author (Year) | Overall (32 Studies) | | Spirometry (20 Studies) | | Respiratory Symptoms (16 Studies) | | Radiology (8 Studies) | |
|---|---|---|---|---|---|---|---|---|---|
| | | n/N | Prevalence Percent (95% CI) | n/N | Prevalence Percent (95% CI) | n/N | Prevalence Percent (95% CI) | n/N | Prevalence) Percent (95% CI |
| 1 | Plit M (1998) | 39/74 | 52.7 (41.3–64.1) | 39/74 | 52.7 (41.3–64.1) | | | | |
| 2 | De Valliere (2004) | 31/33 | 93.9 (85.8–100) | 31/33 | 93.9 (85.8–100) | | | 32/34 | 94.1 (86.2–100) |
| 3 | Ehrlich (2004) | 54/326 | 16.6 (12.5–20.6) | | | 54/326 | 16.6 (12.5–20.6) | | |
| 4 | Muniyandi (2007) | 174/436 | 39.9 (35.3–44.5) | | | 174/436 | 39.9 (35.3–44.5) | | |
| 5 | Cruz R (2008) | 64/96 | 66.7 (57.2–76.1) | 64/96 | 66.7 (57.2–76.1) | | | 86/96 | 89.6 (83.5–95.7) |
| 6 | Maguire (2009) | 17/69 | 24.6 (14.5–34.8) | 17/69 | 24.6 (14.5–34.8) | | | | |
| 7 | Banu Rekha (2009) | 96/148 | 64.9 (57.2–72.6) | 96/148 | 64.9 (57.2–72.6) | 58/198 | 29.3 (23.0–35.6) | 170/198 | 85.9 (81.0–90.7) |
| 8 | Idolor (2011) | 15/26 | 57.7 (38.7–76.7) | | | 15/26 | 57.7 (38.7–76.7) | | |
| 9 | Akkara (2013) | 223/257 | 86.8 (82.6–90.9) | 223/257 | 86.8 (82.6–90.9) | 224/264 | 84.8 (80.5–89.2) | 145/264 | 54.9 (48.9–60.9) |
| 10 | Perfura-Yone (2014) | 110/177 | 62.1 (55.0–69.3) | 67/177 | 37.9 (30.7–45.0) | 110/177 | 62.1 (55.0–69.3) | | |
| 11 | Akanbi (2015) | 15/82 | 18.3 (9.9–26.7) | 15/82 | 18.3 (9.9–26.7) | | | | |
| 12 | Binegdie (2015) | 134/725 | 18.5 (15.7–21.3) | | | | | 134/725 | 18.5 (15.7–21.3) |
| 13 | De la Mora (2015) | 24/70 | 34.3 (23.2–45.4) | 24/70 | 34.3 (23.2–45.4) | | | | |
| 14 | Menon (2015) | 178/441 | 40.4 (35.8–44.9) | | | | | 178/441 | 40.4 (35.8–44.9) |
| 15 | Mbatchou Ngahane (2016) | 122/269 | 45.4 (39.4–51.3) | 122/269 | 45.4 (39.4–51.3) | | | | |
| 16 | Snène H (2016) | 6/33 | 18.2 (5.0–31.3) | | | 6/33 | 18.2 (5.0–31.3) | | |
| 17 | Attia (2018) | 35/96 | 36.5 (26.8–46.1) | 35/96 | 36.5 (26.8–46.1) | 52/96 | 54.2 (44.2–64.1) | | |
| 18 | Katoto (2018) | 247/441 | 56.0 (51.4–60.6) | | | 247/441 | 56.0 (51.4–60.6) | | |
| 19 | Magitta (2018) | 9/18 | 50.0 (26.9–73.1) | 9/18 | 50.0 (26.9–73.1) | | | | |
| 20 | North (2018) | 6/18 | 33.3 (11.6–55.1) | 6/18 | 33.3 (11.6–55.1) | | | | |
| 21 | Singla (2018) | 41/42 | 97.6 (93.0–100) | 41/42 | 97.6 (93.0–100) | 44/46 | 95.7 (89.8–100) | 46/46 | 100 (100–100) |
| 22 | Van Kampen (2019) | 326/798 | 40.9 (37.4–44.3) | | | 124/798 | 15.5 (13.0–18.1) | 326/798 | 40.9 (37.6–44.3) |
| 23 | Chin (2019) | 58/162 | 35.8 (28.4–43.2) | 58/162 | 35.8 (28.4–43.2) | 59/175 | 33.5 (26.5–40.5) | 96/116 | 82.8 (75.9–89.6) |
| 24 | Fiogbe A (2019) | 85/189 | 45.0 (37.9–52.1) | 85/189 | 45.0 (37.9–52.1) | | | | |
| 25 | Gupte (2019) | 132/172 | 76.7 (70.4–83.1) | 132/172 | 75.7 (70.4–83.1) | | | | |
| 26 | Osman (2019) | 28/51 | 54.9 (41.2–68.6) | | | 28/51 | 54.9 (41.2–68.6) | | |
| 27 | Kayongo (2020) | 14/66 | 21.2 (11.3–31.1) | 14/66 | 21.2 (11.3–31.1) | | | | |
| 28 | Meghji (2020) | 125/365 | 34.2 (29.4–39.1) | 125/365 | 34.2 (29.4–39.1) | 246/405 | 60.7 (56.0–65.5) | | |
| 29 | Khosa (2020) | 40/62 | 64.5 (52.6–76.4) | 40/62 | 64.5 (52.6–76.4) | | | | |
| 30 | Auld (2021) | 30/92 | 32.6 (23.0–42.2) | 30/92 | 32.6 (23.0–42.2) | 12/93 | 12.9 (6.1–19.7) | | |
| 31 | Ddungu (2021) | 7/72 | 9.7 (2.9–16.6) | 7/72 | 9.7 (2.9–16.6) | 26/72 | 36.1 (25.0–47.2) | | |
| 32 | Mpagama (2021) | 146/219 | 66.7 (60.4–72.9) | 146/219 | 66.7 (60.4–72.9) | 98/219 | 44.7 (38.2–51.3) | 177/219 | 80.8 (75.6–86.0) |

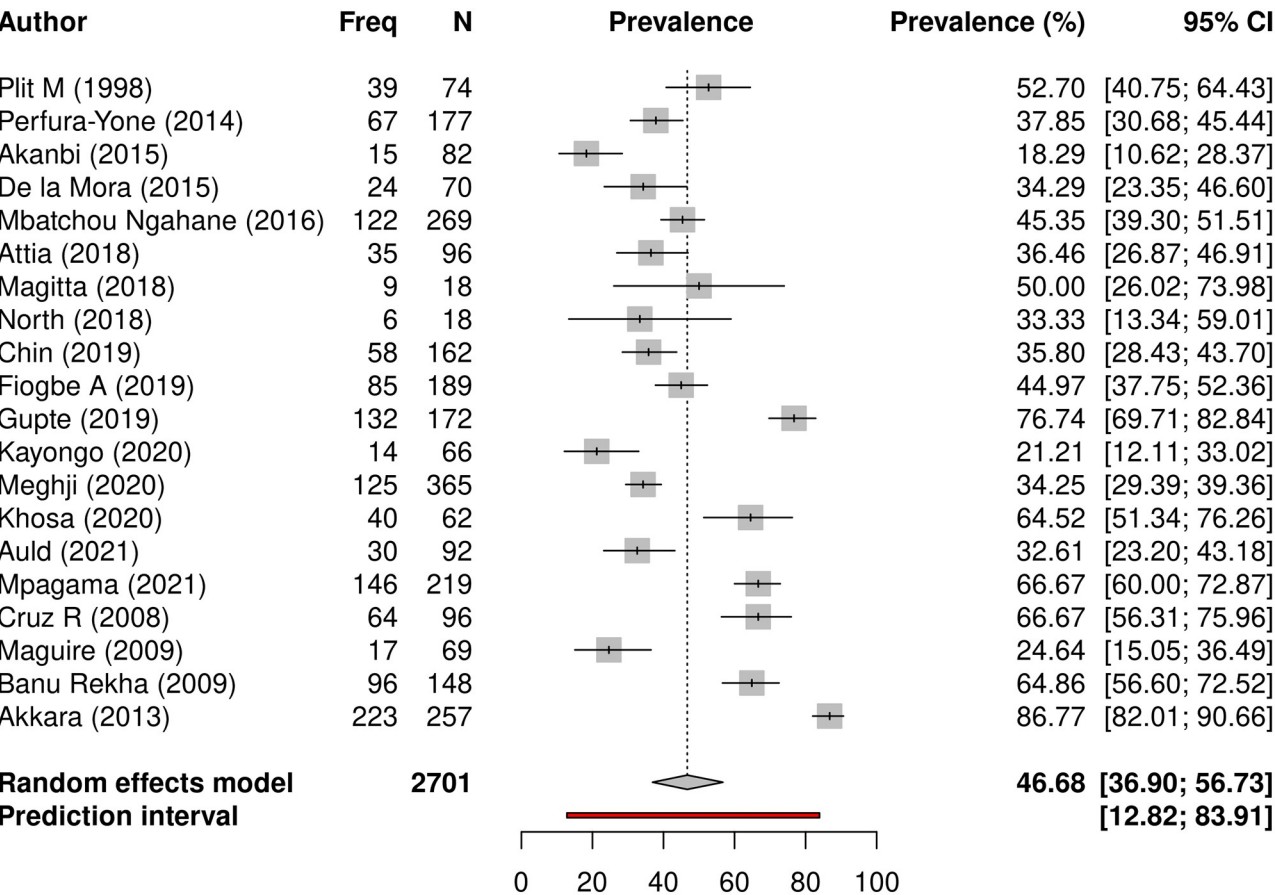

| Author | Freq | N | Prevalence | Prevalence (%) | 95% CI |
|---|---|---|---|---|---|
| Plit M (1998) | 39 | 74 | | 52.70 | [40.75; 64.43] |
| Perfura-Yone (2014) | 67 | 177 | | 37.85 | [30.68; 45.44] |
| Akanbi (2015) | 15 | 82 | | 18.29 | [10.62; 28.37] |
| De la Mora (2015) | 24 | 70 | | 34.29 | [23.35; 46.60] |
| Mbatchou Ngahane (2016) | 122 | 269 | | 45.35 | [39.30; 51.51] |
| Attia (2018) | 35 | 96 | | 36.46 | [26.87; 46.91] |
| Magitta (2018) | 9 | 18 | | 50.00 | [26.02; 73.98] |
| North (2018) | 6 | 18 | | 33.33 | [13.34; 59.01] |
| Chin (2019) | 58 | 162 | | 35.80 | [28.43; 43.70] |
| Fiogbe A (2019) | 85 | 189 | | 44.97 | [37.75; 52.36] |
| Gupte (2019) | 132 | 172 | | 76.74 | [69.71; 82.84] |
| Kayongo (2020) | 14 | 66 | | 21.21 | [12.11; 33.02] |
| Meghji (2020) | 125 | 365 | | 34.25 | [29.39; 39.36] |
| Khosa (2020) | 40 | 62 | | 64.52 | [51.34; 76.26] |
| Auld (2021) | 30 | 92 | | 32.61 | [23.20; 43.18] |
| Mpagama (2021) | 146 | 219 | | 66.67 | [60.00; 72.87] |
| Cruz R (2008) | 64 | 96 | | 66.67 | [56.31; 75.96] |
| Maguire (2009) | 17 | 69 | | 24.64 | [15.05; 36.49] |
| Banu Rekha (2009) | 96 | 148 | | 64.86 | [56.60; 72.52] |
| Akkara (2013) | 223 | 257 | | 86.77 | [82.01; 90.66] |
| **Random effects model** | | **2701** | | **46.68** | **[36.90; 56.73]** |
| **Prediction interval** | | | | | **[12.82; 83.91]** |

Heterogeneity: $I^2 = 95\%$, $\tau^2 = 0.6843$, $\chi^2_{19} = 348.49$ ($p < 0.01$)

**Fig 2. Prevalence of abnormal lung function following TB.** A forest plot showing individual study prevalence and pooled prevalence of residual impairment of lung function in 20 studies that conducted spirometry in a pooled sample size of 2701 study participants post-TB.

Eight out of the 30 studies assessed both symptom burden and lung function in their study participants. In order to assess whether the two measures identify similar or differing levels of post-TB morbidity we developed a forest plot to compare the two outcomes for participants from the subset of eight studies. The pooled prevalence of abnormal spirometry was 50.6% (32.8–68.2%), and symptoms was 47.6% (28.1–67.8%), with no significant difference between the measures (p = 0.80) (S1 Fig).

**Prevalence of radiologic structural lung abnormalities following TB.** Persisting structural abnormalities from chest radiology after completion of TB treatment was reported in 2857 patients from 8 studies, and 1316 (46.1%) had abnormal chest imaging. Prevalence of abnormal radiology findings after completion of TB treatment varied across individual studies and ranged between 18.5 and 89.6% (Table 2). Meta-analysis revealed the pooled prevalence to be 64.6% (95% CI: 39.5–83.6%), with significant heterogeneity across studies ($I^2 = 99\%$, t2 = 1.4776, p<0.01) (Fig 4).

**Overall prevalence of PTLD as assessed by either abnormal lung function, persisting respiratory symptoms or radiology.** In order to provide a model that would give insight on prevalence of PTLD across all 30 studies, where studies used two or three measures to evaluate

| Author | Freq | N | Prevalence | Prevalence (%) | 95% CI |
|---|---|---|---|---|---|
| Perfura-Yone (2014) | 110 | 177 | | 62.15 | [54.56; 69.32] |
| Snene H (2016) | 6 | 33 | | 18.18 | [ 6.98; 35.46] |
| Attia (2018) | 52 | 96 | | 54.17 | [43.69; 64.38] |
| Katoto (2018) | 247 | 441 | | 56.01 | [51.24; 60.70] |
| Van Kampen (2019) | 124 | 798 | | 15.54 | [13.09; 18.24] |
| Chin (2019) | 59 | 176 | | 33.52 | [26.60; 41.01] |
| Osman (2019) | 28 | 51 | | 54.90 | [40.34; 68.87] |
| Meghji (2020) | 246 | 405 | | 60.74 | [55.80; 65.53] |
| Ehrlich (2004) | 54 | 326 | | 16.56 | [12.70; 21.05] |
| Auld (2021) | 12 | 93 | | 12.90 | [ 6.85; 21.45] |
| Ddungu (2021) | 26 | 72 | | 36.11 | [25.12; 48.29] |
| Mpagama (2021) | 98 | 219 | | 44.75 | [38.05; 51.60] |
| Muniyandi (2007) | 174 | 436 | | 39.91 | [35.28; 44.68] |
| Banu Rekha (2009) | 58 | 198 | | 29.29 | [23.06; 36.16] |
| Idolor (2011) | 15 | 26 | | 57.69 | [36.92; 76.65] |
| Akkara (2013) | 224 | 264 | | 84.85 | [79.94; 88.95] |
| **Random effects model** | | **3811** | | **41.00** | **[29.39; 53.70]** |
| **Prediction interval** | | | | | **[ 8.03; 84.68]** |

Heterogeneity: $I^2$ = 97%, $\tau^2$ = 0.8773, $\chi^2_{15}$ = 597.40 ($p$ < 0.01)

**Fig 3. Prevalence of persistent respiratory symptoms following TB.** A forest plot showing individual study prevalence and pooled prevalence of persisting respiratory symptoms in 16 studies that evaluated a pooled sample size of 3811 study participants for residual symptoms post-TB.

| Author | Freq | N | Prevalence | Prevalence (%) | 95% CI |
|---|---|---|---|---|---|
| Binegdie (2015) | 134 | 725 | | 18.48 | [15.72; 21.50] |
| Menon (2015) | 178 | 441 | | 40.36 | [35.75; 45.11] |
| Van Kampen (2019) | 326 | 798 | | 40.85 | [37.42; 44.35] |
| Chin (2019) | 96 | 116 | | 82.76 | [74.64; 89.14] |
| Mpagama (2021) | 177 | 219 | | 80.82 | [74.97; 85.81] |
| Cruz R (2008) | 86 | 96 | | 89.58 | [81.68; 94.89] |
| Banu Rekha (2009) | 170 | 198 | | 85.86 | [80.21; 90.39] |
| Akkara (2013) | 145 | 264 | | 54.92 | [48.71; 61.03] |
| **Random effects model** | | **2857** | | **64.63** | **[39.53; 83.63]** |
| **Prediction interval** | | | | | **[ 7.20; 97.73]** |

Heterogeneity: $I^2$ = 99%, $\tau^2$ = 1.4776, $\chi^2_7$ = 488.49 ($p$ < 0.01)

**Fig 4. Prevalence of radiologic structural lung abnormalities following TB.** A forest plot showing individual study prevalence and pooled prevalence of abnormal chest radiology in 8 studies that conducted chest imaging in a pooled sample of 2857 study participants post-TB.

for post-TB sequelae, we selected the spirometry prevalence value where available (n = 7), and clinical symptoms if no spirometry was available (n = 1). In studies that used only one measure, that value was adopted to represent PTLD for the patients in the given study. The rationale for this approach was spirometry is an objective measure of physiologic lung function and identifies milder forms of impairment that may not yet manifest as obvious clinical symptoms in the patient. On the other hand, what is more concerning to a patient is persisting symptoms, which often affect their quality of life, and drive their health care seeking decisions. Using this approach, data was available for 6050 patients across the 30 studies, and 2559 (42.3%) had abnormal lung function or persisting respiratory symptoms or abnormal radiology. Meta-analysis revealed the overall pooled prevalence of PTLD from the 30 studies to be 42.7% (95% CI 34.6–51.1%), with significant heterogeneity across studies ($I^2$ = 96%, t2 = 0.7710, p<0.01) (Fig 5).

We examined data from the subset of six studies that enrolled some patients who had received retreatment for TB along with first time TB patients, and found that retreatment patients comprised a minority of the study population in five of the six studies [25, 28, 36, 41, 44, 50]. However, one study [41] exclusively enrolled retreatment TB patients. In these mixed studies, PTLD prevalence from spirometry or clinical symptoms ranged from 34.2% to 66.7%.

**Prevalence of PTLD among study participants with prior multi-drug resistant tuberculosis.** Sequelae following MDR-TB are reported in two studies involving 80 study participants. Both studies conducted spirometry and reported high prevalence of impaired lung function of 93.9% (31/33 patients) [20] and 97.6% (41/42 patients) [39] respectively (Table 2). One study assessed and reported high prevalence of persisting respiratory symptoms in MDR TB survivors at 95.7% (44/46) [39]. Both studies conducted chest radiography among MDR TB survivors and reported high prevalence of post-treatment radiologic abnormalities of 94.1% (32/34 participants) [20] and 100.0% (42/42 participants) [39] respectively following recovery from MDR TB.

## Factors associated with post-TB lung disease

Sub-group meta-analyses were conducted separately to evaluate association between specific factors and the outcome persistent respiratory symptoms (16 studies), abnormal spirometry (20 studies), and finally using composite outcome to enable insight using all 30 studies (spirometry, or symptoms or radiology). Data from the 30 studies of participants with prior clinically diagnosed or confirmed drug-susceptible TB was included in these subgroup meta-analyses, and random effects models were developed to assess for significant difference in pooled prevalence between subgroups.

**Factors associated with persisting respiratory symptoms.** Analyses were conducted using prevalence of persisting symptoms as the outcome measure from 16 studies to evaluate for associated factors, results are displayed in Table 3. Persistent respiratory symptoms were more prevalent in rural settings compared to urban or mixed settings (68.8% vs 39.1% vs 27.2% respectively, p = 0.0035) (Fig 6). Symptom prevalence was higher in SE Asia than Africa (S2a Fig), higher in low- and lower-middle income compared to upper-middle income countries (S2b Fig), and in adults compared to children, but differences were not statistically significant for these factors (S2c Fig).

**Factors associated with abnormal lung function.** We examined factors associated with abnormal lung function (spirometry) in the 20 studies that assessed lung function, the results are summarised in Table 4. Abnormal lung function was most prevalent in mixed rural-urban settings, followed by rural settings, and lowest in urban settings (71.6%% vs 49.8% vs 41.9% respectively, p<0.0001) (S3c Fig).

| Author | Freq | N | Prevalence | Prevalence (%) | 95% CI |
|--------|------|---|------------|----------------|--------|
| Plit M (1998) | 39 | 74 | | 52.70 | [40.75; 64.43] |
| Perfura-Yone (2014) | 110 | 177 | | 62.15 | [54.56; 69.32] |
| Akanbi (2015) | 15 | 82 | | 18.29 | [10.62; 28.37] |
| Binegdie (2015) | 134 | 725 | | 18.48 | [15.72; 21.50] |
| De la Mora (2015) | 24 | 70 | | 34.29 | [23.35; 46.60] |
| Menon (2015) | 178 | 441 | | 40.36 | [35.75; 45.11] |
| Mbatchou Ngahane (2016) | 122 | 269 | | 45.35 | [39.30; 51.51] |
| Snene H (2016) | 6 | 33 | | 18.18 | [ 6.98; 35.46] |
| Attia (2018) | 35 | 96 | | 36.46 | [26.87; 46.91] |
| Katoto (2018) | 247 | 441 | | 56.01 | [51.24; 60.70] |
| Magitta (2018) | 9 | 18 | | 50.00 | [26.02; 73.98] |
| North (2018) | 6 | 18 | | 33.33 | [13.34; 59.01] |
| Van Kampen (2019) | 326 | 798 | | 40.85 | [37.42; 44.35] |
| Chin (2019) | 58 | 162 | | 35.80 | [28.43; 43.70] |
| Fiogbe A (2019) | 85 | 189 | | 44.97 | [37.75; 52.36] |
| Gupte (2019) | 132 | 172 | | 76.74 | [69.71; 82.84] |
| Osman (2019) | 28 | 51 | | 54.90 | [40.34; 68.87] |
| Kayongo (2020) | 14 | 66 | | 21.21 | [12.11; 33.02] |
| Meghji (2020) | 125 | 365 | | 34.25 | [29.39; 39.36] |
| Khosa (2020) | 40 | 62 | | 64.52 | [51.34; 76.26] |
| Ehrlich (2004) | 54 | 326 | | 16.56 | [12.70; 21.05] |
| Auld (2021) | 30 | 92 | | 32.61 | [23.20; 43.18] |
| Ddungu (2021) | 7 | 72 | | 9.72 | [ 4.00; 19.01] |
| Mpagama (2021) | 146 | 219 | | 66.67 | [60.00; 72.87] |
| Muniyandi (2007) | 174 | 436 | | 39.91 | [35.28; 44.68] |
| Cruz R (2008) | 64 | 96 | | 66.67 | [56.31; 75.96] |
| Maguire (2009) | 17 | 69 | | 24.64 | [15.05; 36.49] |
| Banu Rekha (2009) | 96 | 148 | | 64.86 | [56.60; 72.52] |
| Idolor (2011) | 15 | 26 | | 57.69 | [36.92; 76.65] |
| Akkara (2013) | 223 | 257 | | 86.77 | [82.01; 90.66] |
| **Random effects model** | | **6050** | | **42.69** | **[34.64; 51.14]** |
| **Prediction interval** | | | | | **[10.67; 82.29]** |

0    20    40    60    80    100

Heterogeneity: $I^2 = 96\%$, $\tau^2 = 0.7710$, $\chi^2_{29} = 709.12$ ($p < 0.01$)

**Fig 5. Overall prevalence of PTLD.** A forest plot showing the overall prevalence of PTLD from all 30 studies involving a pooled sample size of 6050 study participants. Studies used varying methods of post-TB assessment, for this composite model if a study used more than one method, spirometry data was adopted, symptom data if no spirometry was done, and radiology data if neither spirometry nor symptom assessment was done.

Abnormal lung function was most prevalent in SE Asia, followed by the Americas and Africa (65.9 vs 50.8 vs 40.5% respectively, p = 0.154) (S3a Fig). We noted very wide 95% CI for prevalence estimates in the Americas which had only two studies and 166 participants, so we developed a second model without the Americas, and there appeared to be a trend for increased pooled prevalence of abnormal spirometry in SE Asia compared to Africa (65.92% vs 40.48% respectively, p = 0.07). Prevalence of abnormal lung function was similar across income settings and age group (p>0.05), (Table 4, S3b and S3d Fig).

**Table 3. Factors associated with persisting respiratory symptoms following TB.**

| Factor | Subgroup | No. studies | No. participants | Pooled Prevalence Percent (95% CI) | Heterogeneity I² (%) | Test subgroup differences. P |
|---|---|---|---|---|---|---|
| **WHO Region**[a] | Africa | 11 | 2,854 | 38.6 (26.0–52.8) | 98 | 0.3892 |
| | SE Asia | 3 | 898 | 53.5 (6.4–95.1) | 99 | |
| | *Overall* | *14* | *3,752* | *41.7 (29.1–55.5)* | *98* | |
| **Income Level** | LIC | 4 | 1,716 | 40.2 (14.7–72.4) | 99 | 0.2348 |
| | LMIC | 9 | 1,625 | 47.7 (31.8–64.0) | 96 | |
| | UMIC | 3 | 470 | 24.4 (3.1–76.5) | 95 | |
| | *Overall* | *16* | *3,811* | *41.0 (29.4–53.7)* | *98* | |
| **Urban-Rural** | Urban | 9 | 1,301 | 39.1 (25.7–54.4) | 94 | 0.0035 |
| | Mixed | 4 | 1,779 | 27.2 (11.3–52.4) | 98 | |
| | Rural | 3 | 731 | 68.8 (27.3–92.9) | 97 | |
| | *Overall* | *16* | *3,811* | *41.0 (29.4–53.7)* | *97* | |
| **Age group** | Child only | 2 | 129 | 35.4 (0.03–99.9) | 91 | 0.5983 |
| | Adolescents & adults | 4 | 1,970 | 34.1 (9.5–71.9) | 99 | |
| | Adults only | 8 | 1,100 | 47.5 (27.9–67.9) | 96 | |
| | *Overall* | *14* | *3,199* | *41.6 (28.3–56.4)* | *98* | |

A table showing the relationship between geographic distribution, country income level, urbanisation participant age group and experiencing persisting respiratory symptoms post TB infection and treatment.

[a] No studies with symptom outcome data from the Americas, and only one study in Eastern Mediterranean of small sample size of 33 participants reported on post-TB sequelae.

**Factors associated with overall PTLD using composite outcome—abnormal spirometry or persistent symptoms or abnormal radiology.** To incorporate maximal data for factor analysis of occurrence of any type of lung sequelae across all 30 studies, we developed models using composite PTLD outcome data (either abnormal spirometry or persisting symptoms or abnormal radiology as available from each of the 30 studies). In studies that reported more than one measure, spirometry prevalence was adopted, followed by symptom prevalence if no spirometry done, and radiology prevalence in the two studies that only provided radiology outcomes.

PTLD varied significantly across WHO regions, from greatest to lowest prevalence as follows: highest in the Western Pacific and SE Asian regions (57.7% and 57.5% respectively), followed by the Americas (50.8%), then the African and Eastern Mediterranean regions (38.2% and 18.2% respectively, p = 0.0118). Of note, the Eastern Mediterranean region had only one study from Tunisia with a small sample size of 33. (Table 5 and Fig 7)

The composite outcome meta-analysis model compared HIV status data for 1599 participants from 17 studies. HIV uninfected study participants appeared to have a significantly higher prevalence of PTLD than HIV infected participants (66.9% vs 32.8% respectively, p = 0.0013), (Table 5 and Fig 8).

Age-group analysis revealed increasing prevalence of PTLD with age (children 29.3% vs mixed adolescent/adult populations 38.0% vs adult studies 46.9% respectively), however this difference was not statistically significant (p = 0.1681). Prevalence of any type of PTLD using composite outcome was similar across income settings, urban-rural settings, and sex (p>0.05) (S4a, S4b, S4c and S4d Fig).

Prior MDR-TB participants appeared to have higher prevalence of PTLD exceeding 90% in the two studies, compared to overall PTLD prevalence of 42.7% from the 30 studies involving

| Author | Freq | N | Prevalence | Prevalence (%) | 95% CI |
|---|---|---|---|---|---|
| **urban_or_rural = Urban** | | | | | |
| Perfura-Yone (2014) | 110 | 177 | | 62.15 | [54.56; 69.32] |
| Snene H (2016) | 6 | 33 | | 18.18 | [ 6.98; 35.46] |
| Attia (2018) | 52 | 96 | | 54.17 | [43.69; 64.38] |
| Chin (2019) | 59 | 176 | | 33.52 | [26.60; 41.01] |
| Osman (2019) | 28 | 51 | | 54.90 | [40.34; 68.87] |
| Meghji (2020) | 246 | 405 | | 60.74 | [55.80; 65.53] |
| Auld (2021) | 12 | 93 | | 12.90 | [ 6.85; 21.45] |
| Ddungu (2021) | 26 | 72 | | 36.11 | [25.12; 48.29] |
| Banu Rekha (2009) | 58 | 198 | | 29.29 | [23.06; 36.16] |
| **Random effects model** | | **1301** | | **39.09** | **[25.70; 54.35]** |
| Heterogeneity: $I^2 = 94\%$, $\tau^2 = 0.5885$, $\chi^2_8 = 131.38$ ($p < 0.01$) | | | | | |
| | | | | | |
| **urban_or_rural = Mixed** | | | | | |
| Van Kampen (2019) | 124 | 798 | | 15.54 | [13.09; 18.24] |
| Ehrlich (2004) | 54 | 326 | | 16.56 | [12.70; 21.05] |
| Mpagama (2021) | 98 | 219 | | 44.75 | [38.05; 51.60] |
| Muniyandi (2007) | 174 | 436 | | 39.91 | [35.28; 44.68] |
| **Random effects model** | | **1779** | | **27.19** | **[11.26; 52.35]** |
| Heterogeneity: $I^2 = 98\%$, $\tau^2 = 0.4450$, $\chi^2_3 = 137.58$ ($p < 0.01$) | | | | | |
| | | | | | |
| **urban_or_rural = Rural** | | | | | |
| Katoto (2018) | 247 | 441 | | 56.01 | [51.24; 60.70] |
| Idolor (2011) | 15 | 26 | | 57.69 | [36.92; 76.65] |
| Akkara (2013) | 224 | 264 | | 84.85 | [79.94; 88.95] |
| **Random effects model** | | **731** | | **68.81** | **[27.26; 92.85]** |
| Heterogeneity: $I^2 = 97\%$, $\tau^2 = 0.4491$, $\chi^2_2 = 57.23$ ($p < 0.01$) | | | | | |
| | | | | | |
| **Random effects model** | | **3811** | | **41.00** | **[29.39; 53.70]** |
| **Prediction interval** | | | | | **[ 8.03; 84.68]** |

0  20  40  60  80  100

Heterogeneity: $I^2 = 97\%$, $\tau^2 = 0.8773$, $\chi^2_{15} = 597.40$ ($p < 0.01$)
Test for subgroup differences: $\chi^2_2 = 11.30$, df = 2 ($p < 0.01$)

**Fig 6. Prevalence of persisting respiratory symptoms in urban, mixed, and rural settings.** A forest plot showing subgroup meta-analyses comparing prevalence of persisting symptoms following TB treatment across rural versus mixed versus urban settings.

prior clinically diagnosed or drug-susceptible TB participants, and 95% CI for these prevalence estimates were not overlapping. However, due to the small sample size of prior MDR-TB participants no meta-analysis was attempted to compare the two risk groups.

All papers were reviewed for data on biomass fuel exposure / household air pollution as a modifiable risk factor, however only three studies sought information on the same, therefore data was inadequate to assess this as a risk factor across the studies.

**Table 4. Factors associated with residual abnormal lung function following TB.**

| Factor | Subgroup | No. studies | No. participants | Pooled Prevalence Percent (95% CI) | Heterogeneity I² (%) | Test subgroup differences. P |
|---|---|---|---|---|---|---|
| **WHO Region** [b] | Africa | 14 | 1889 | 40.5 (32.4–49.1) | 89 | 0.1545 |
| | Americas | 2 | 166 | 50.8 (0.2–99.8) | 94 | |
| | SE Asia | 4 | 646 | 65.9 (25.0–91.8) | 97 | |
| | *Overall* | *20* | *2701* | *46.7 (36.9–56.7)* | *95* | |
| **Income Level** | LIC | 4 | 511 | 37.6 (16.0–65.6) | 89 | 0.5430 |
| | LMIC | 12 | 1858 | 49.7 (35.3–64.2) | 96 | |
| | UMIC | 4 | 332 | 46.5 (25.3–69.1) | 89 | |
| | *Overall* | *20* | *2701* | *46.7 (36.9–56.7)* | *95* | |
| **Urban-Rural** | Urban | 14 | 1951 | 41.9 (33.6–50.6) | 89 | <0.0001 |
| | Mixed | 2 | 391 | 71.6 (21.0–96.0) | 79 | |
| | Rural | 4 | 359 | 49.8 (11.5–88.4) | 97 | |
| | *Overall* | *20* | *2701* | *46.7 (36.9–56.7)* | *95* | |
| **Age Group** | Child only | 1 | 96 | 36.5 (26.9–46.9) | - | 0.2620 |
| | Adolescents & adults | 3 | 730 | 48.1 (19.3–78.3) | 94 | |
| | Adults only | 15 | 1713 | 47.7 (35.0–60.8) | 95 | |
| | *Overall* | *19* | *2539* | *47.3 (37.0–57.8)* | *95* | |

A table showing the relationship between geographic distribution, country income level, urbanisation, participant age group and residual abnormal lung function as assessed by spirometry.

[b] The study from Eastern Mediterranean did not do spirometry, and one study from Western Pacific of small sample size of 26 participants reported on spirometry.

## Discussion

This systematic review and meta-analysis identified an increase in research focusing on sequelae among former TB patients, with half of the 32 identified publications released over the past five years, two-thirds of studies from the African region, and half of the papers providing insight into persons living with HIV who experienced prior TB. The review and meta-analysis quantitatively reveal a high burden of post-TB lung disease among persons living in low- and middle-income countries, with pooled prevalence of persistent respiratory symptoms of 41.0%, of abnormal spirometry 46.7%, and of radiologic structural abnormalities somewhat higher at 64.6%. Studies on children were lacking, and young children below 6 years were completely excluded from all identified studies.

This meta-analysis reveals that approximately half of persons surviving TB have impaired lung function, which appears higher than that found in the meta-analysis by Fan et al, who reported 21% with obstructive airway disease from 16 studies [9]. These differences may be due to inclusion of several studies from high-income countries in both reviews, where patients have better access to health care, may be diagnosed at earlier stages of TB disease with less severe lung pathology, and generally have lower exposure to biomass smoke than those living in low-income settings, which is likely to contribute to COPD. Both Fan et al and Byrne et al focused only on post-TB obstructive airway disease (COPD), whereas several of our studies reported on all types of spirometric abnormality–obstructive as well as restrictive patterns thereby providing a more comprehensive picture of the burden of functional impairment in the study populations. We also provide valuable insight into post-TB functional impairment in individuals living in low-income countries.

We found that the burden of chronic respiratory symptoms persisting after completion of TB treatment was high at 42.6%, and these data emerged largely from studies done within the

**Table 5. Factors associated with post-TB lung disease using composite outcome.**

| Factor | Subgroup | No. studies | No. participants | Pooled Prevalence Percent (95% CI) | Heterogeneity I² (%) | Test subgroup differences. P |
|---|---|---|---|---|---|---|
| **HIV status** | HIV negative | 6 | 788 | 66.9 (43.2–84.3) | 96 | 0.0013 |
| | HIV positive | 11 | 811 | 32.8 (22.8–44.6) | 82 | |
| | *Overall* | *17* | *1599* | *44.8 (31.9–58.4)* | *94* | |
| **Sex** | Female | 5 | 499 | 47.0 (21.0–74.7) | 93 | 0.5650 |
| | Male | 5 | 695 | 39.3 (20.2–62.3) | 94 | |
| | *Overall* | *10* | *1194* | *42.8 (28.8–58.1)* | *93* | |
| **Age group** | Child only | 2 | 129 | 29.3 (0.6–96.8) | 72 | 0.1681 |
| | Adolescents/Adults mixed | 7 | 3020 | 38.0 (23.0–55.8) | 98 | |
| | Adults only | 19 | 2303 | 46.9 (35.4–58.7) | 95 | |
| | *Overall* | *28* | *5452* | *43.0 (34.4–52.1)* | *96* | |
| **WHO Region** | Eastern Mediterranean | 1 | 33 | 18.2 (7.0–35.5) | - | 0.0118 |
| | Africa | 20 | 4302 | 38.2 (29.9–47.2) | 95 | |
| | Americas | 2 | 166 | 50.8 (0.2–99.8) | 94 | |
| | SE Asia | 6 | 1523 | 57.5 (31.4–79.9) | 98 | |
| | Western Pacific | 1 | 26 | 57.7 (36.9–76.7) | NA | |
| | *Overall* | *30* | *6050* | *42.7 (34.6–51.1)* | *96* | |
| **Income Level** | LIC | 8 | 2547 | 32.7 (19.1–50.0) | 97 | 0.2180 |
| | LMIC | 16 | 2794 | 48.4 (36.8–60.3) | 95 | |
| | UMIC | 6 | 709 | 41.4 (23.8–61.6) | 95 | |
| | *Overall* | *30* | *6050* | *42.7 (34.6–51.1)* | *96* | |
| **Urban-Rural** | Urban | 19 | 3273 | 38.7 (30.2–48.0) | 94 | 0.4079 |
| | Mixed | 5 | 1951 | 47.4 (21.2–75.1) | 98 | |
| | Rural | 6 | 826 | 52.6 (26.9–76.9) | 96 | |
| | *Overall* | *30* | *6050* | *42.7 (34.6–51.1)* | *82* | |

A table showing the factors associated with PTLD using composite outcome data from 30 studies (abnormal spirometry or persistent symptoms or abnormal radiology).

last five years, and predominantly from low-income countries. We could find no other published systematic review or meta-analysis on persistent symptom burden following TB, thereby no comparisons of pooled data could be made with prior reviews. This evidence of significant morbidity burden among this vulnerable population has implications for their quality of life and may provide a proxy indicator of what proportion of prior TB patients may seek health care repeatedly even after completion of TB treatment [51].

We noted that although there were more studies evaluating spirometric impairment than those evaluating symptom burden (20 versus 16 studies respectively). In terms of absolute numbers of participants the pooled study populations that provided insight on symptoms were larger than those that conducted spirometry (3811 participants with symptom data versus 2701 with spirometry data), and this was especially seen in studies conducted in low-income countries (1716 symptom data versus 511 spirometry data from LIC studies). We noted that majority of spirometry data was from urban settings, possibly a reflection of poor access to spirometry in rural health facilities. We also noted in studies where both measures were assessed, symptom report was available for the full study population, whereas not all patients were able to do a successful spirometry test.

One important observation was that in the eight studies that reported both symptom burden and lung function in their participants the overall pooled prevalence of any abnormal spirometry was quite similar to the pooled prevalence of persistent symptoms (50.6% vs 47.6%

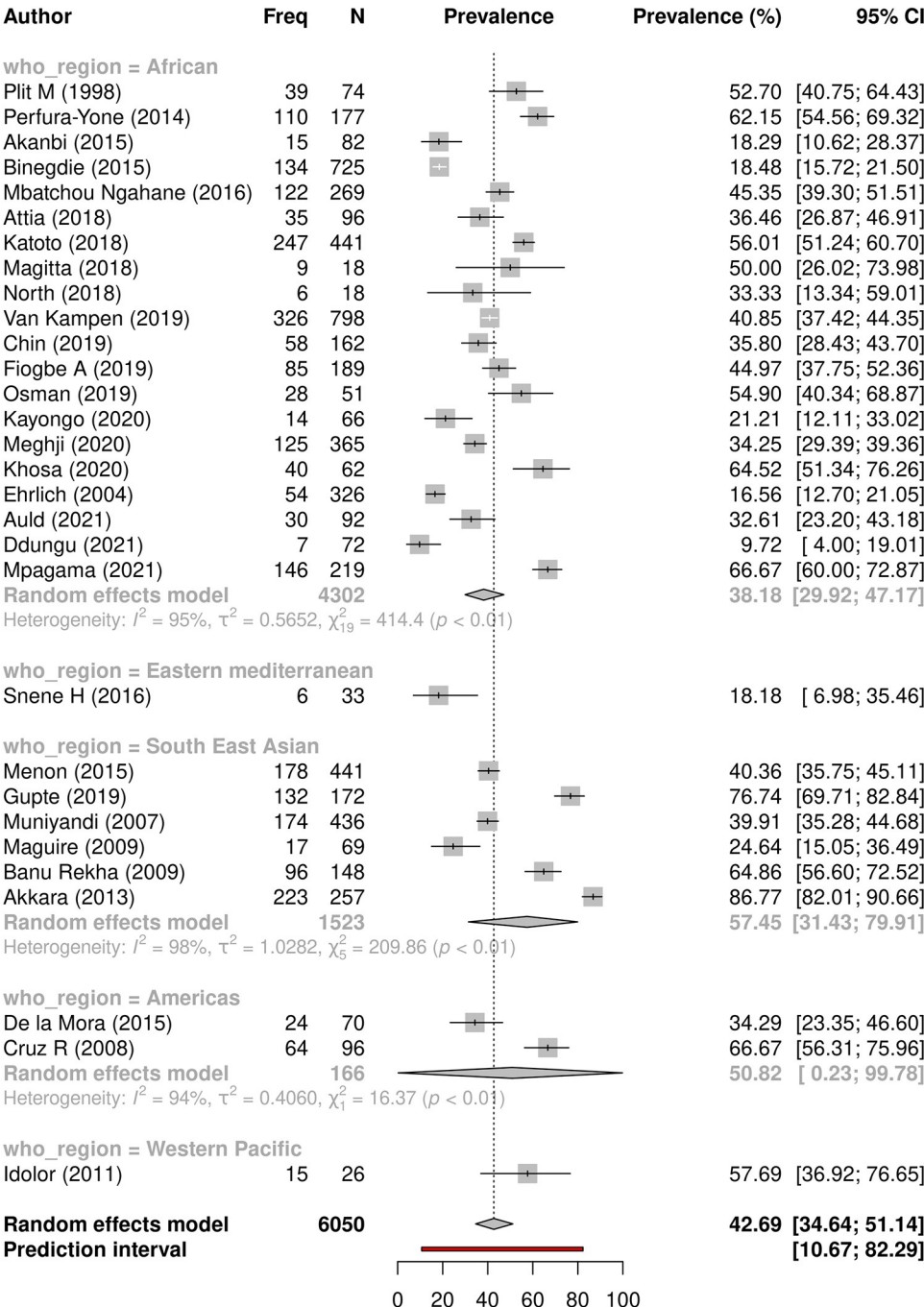

**Fig 7. Prevalence of PTLD across World Health Organisation regions.** A forest plot showing subgroup meta-analysis comparing prevalence of PTLD as a composite outcome (abnormal spirometry or persistent symptoms or abnormal radiology) in all 30 studies across five WHO regions.

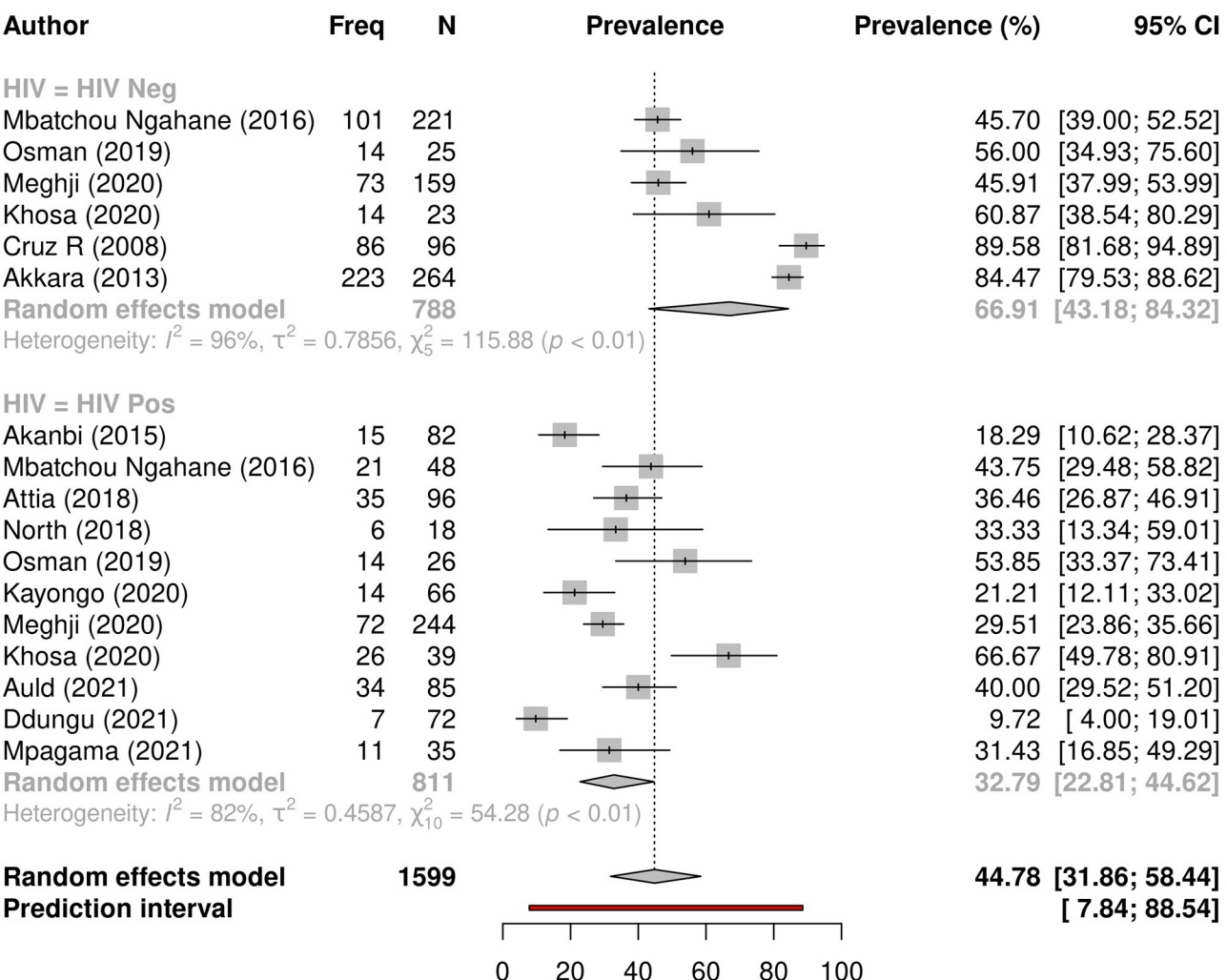

**Fig 8. Prevalence of PTLD comparing HIV uninfected and HIV infected populations.** A forest plot assessing relationship between HIV status and prevalence of PTLD (composite outcome) in studies which documented the HIV status.

respectively), the 95% confidence intervals for these two estimates were overlapping, and statistical tests revealed no significant difference between them (S1 Fig). This suggests that both measures provide similar detection of these patient-centred clinically relevant morbidity measures. One possible implication of this finding is that in resource limited settings with limited access to spirometry, simple screening of patients at the end of TB treatment for persisting respiratory symptoms may be a simple feasible entry point to identification of individuals with post-TB sequelae, who may then be linked into further evaluation and treatment for sequelae and pulmonary rehabilitation as needed.

In the few studies that conducted imaging combined with a second measure of PTLD in their study participants, prevalence of abnormal imaging was consistently much higher than symptoms or abnormal lung function in the same patients. There was wide variation in how chest radiographs were interpreted and reported, and where information was provided, many

studies included presence of minor structural abnormalities in prevalence of outcome, regardless of whether the minor abnormality is likely to cause clinically significant morbidity. This may explain the wide difference between radiologic prevalence on one hand, and spirometric or symptomatic PTLD prevalence on the other hand, both within individual studies, and from pooled study data in the meta-analyses.

The systematic review by Meghji et al provided a narrative synthesis on imaging defined post-TB lung disease from 27 studies mainly from outside Africa, reporting the range in prevalence of specific imaging abnormalities such as fibrosis or cavitation from individual studies. Their review noted lack of data in persons living with HIV and absence of data from sub-Saharan Africa, and they did not conduct a meta-analysis [10]. Our narrative synthesis augments the previous review, and additionally our meta-analysis provides new and quantitative insight into overall high magnitude of structural lung damage from TB across a wider range of LMICs, and includes new insight in individuals living in sub-Saharan Africa.

## Factors associated with post-TB lung disease

We conducted subgroup meta-analyses for various geographical, economic, and demographic factors against two outcome measures: abnormal spirometry, or persistent respiratory symptoms. Participants living in rural settings appeared to have a significantly higher burden of post-TB symptoms, and functional impairment than those from urban settings. We postulate that various factors could explain this including delay in access to health care and possible more advanced TB at time of diagnosis and treatment initiation, higher exposure to household air pollution from use of unclean cooking fuel such as firewood or charcoal common to rural settings. These have been documented in literature as factors contributing to worse health outcomes in persons living in rural settings [52].

Impaired respiratory function prevalence was compared across WHO geographical regions, and only two regions—SE Asia and the African region had adequate number of studies, whereas the Americas and Eastern Mediterranean regions had two and one study respectively and were therefore omitted from the final sub-group meta-analyses model. Impaired respiratory function was high in both regions but appeared higher in SE Asian study populations (65.9%) than African populations (40.5%). Of note, HIV infected participants (who appear to have lower prevalence of post-TB lung sequelae than HIV uninfected individual) were predominantly represented in African studies which may have contributed to this difference, however further research is required to elucidate what risk factors specific to each region contribute to differences in occurrence of sequelae from TB.

Both HIV infected and uninfected populations had high pooled prevalence of PTLD, however HIV uninfected study populations had two-fold higher burden than HIV infected populations. These data were drawn from 17 studies and appeared consistent across individual studies that included both HIV infected and uninfected individuals. There is evidence that TB-HIV co-infected persons with low CD4 counts are four times more likely to have normal chest radiographs and reduced cavitation than those with higher CD4 counts [53]. Key immune mediators of inflammation that are implicated in lung injury during TB have been shown to be reduced in HIV-TB co-infected individuals compared to HIV-uninfected individuals with TB [6, 54, 55], which may partially explain the reduced occurrence of residual lung sequelae among HIV infected individuals surviving TB. There is strong evidence of increased mortality during and after TB treatment among TB-HIV co-infected individuals, those with more advanced HIV disease are less likely to survive [56, 57]. Half of the studies involving HIV infected participants were cross-sectional and patients were assessed months to years

after completion of TB treatment, and those who died from severe lung injury would not be captured. Studies in HIV infected children have also revealed high burden of chronic lung disease of multifactorial aetiology, and though there is limited data to elucidate what proportion of these lung disease is attributable to TB, evidence suggests that prior TB history is a contributing factor to chronic lung sequelae among HIV infected children and adolescents. A longitudinal cohort study in HIV-infected adolescents demonstrated that prior TB was associated with lower mean lung function parameters compared to those without prior TB [58] and this finding persisted in two-year follow-up assessments [59]. Prevalence of impaired lung function by prior TB exposure status was not reported, making it difficult to make direct comparison with our findings. In a cross-sectional study on HIV infected children and teenagers in Malawi prior TB was not significantly associated with persistent symptoms [60].

Apart from HIV we attempted to examine the magnitude of PTLD by different age-group strata. We found limited disaggregated prevalence data on children and adolescents, with only two studies with relatively few study participants providing data specific to children 6–19 years, and no study examined children below 6 years of age. Where adolescents were included with adults, they were the minority, and disaggregated prevalence of PTLD by age group was not provided. We grouped the studies into child-adolescent only, mixed adolescent-adult studies, and adult only studies, and found pooled prevalence to be lowest in children and highest in adults, and though not statistically significantly different, may suggest that children age above five years survive TB with less residual lung impairment than adults, but requires better paediatric evidence. In recent weeks a cross-sectional study on lung function in children with prior TB from The Gambia was published. Children age below 15 years who had history of previous TB had higher prevalence of impaired lung function compared to controls without prior TB (38.5% versus 17.4% respectively), and chronic cough was associated with impaired lung function [61]. Children age below 5 years were excluded due to inability to perform spirometry. The paucity of studies on young children below five years is a glaring gap and suggests inequity in access to research. Young children contribute a higher proportion of incident TB in the paediatric population and are well known to experience rapid progression to severe disease and are potentially at high risk of lung injury and sequelae [56]. It is unclear whether the lower prevalence of PTLD seen in this review is an underestimate of the true magnitude across the full paediatric population.

The ultimate disaster of post-TB sequelae is disability and death despite surviving the initial disease. Survivors of TB who have clinical symptoms or significant impairment of respiratory function experience life-long morbidity, and those with more severe impairment experience disability and death. A meta-analysis by Romanowski and others revealed 2.92 increased pooled mortality risk (95% CI 2.21–3.84) among persons surviving prior TB compared to those who have never had TB [57]. A recent simulation model of lifetime health outcomes following TB using global epidemiologic data estimates that every incident case of post-TB sequelae experiences 5.8 disability adjusted life-years (DALYs), one third of total DALYs accrue 15 or more years after the active TB disease, and this figure is higher for younger persons, and those living in high TB incident settings [51].

**Limitations & strengths.**   The majority of studies were cross-sectional–which limits insight to evolution of chronic respiratory abnormalities over time and may miss out patients who drop out of follow-up or die (survivor bias), thereby may underestimate actual prevalence. However, this is mitigated by the moderate number of prospective cohort studies (n = 9), which is higher than those reported on in previous systematic reviews on this topic, and many were published after 2017 providing new evidence. Prior TB diagnosis was by self-report in the majority of studies, and due to stigma there is potential for participants to under-report that they have had prior TB.

There was significant heterogeneity for the primary outcome (prevalence of PTLD), however this was an expected finding given the heterogeneity across included studies with regards to study setting, access to health care, study population characteristics and regional differences in TB epidemiology. We therefore applied a random effects model in the meta-analyses to account for heterogeneity and assessed residual heterogeneity using sensitivity analyses.

There are several strengths of this systematic review, our search of existing literature was exhaustive, and identified a large number of studies with a large pooled sample size of more than 6,000 participants from diverse settings including the African region suggesting high generalizability of our findings. We examined three outcome measures of TB sequelae–persisting symptoms, functional impairment, and radiologic abnormalities, and conducted both narrative synthesis as well as meta-analyses of the data, providing a broader insight into various aspects of PTLD than previous systematic reviews have done. The majority of studies were of moderate to high quality and half of the papers were published after 2017 thereby providing a synthesis of new emerging evidence on post-TB sequelae.

## Conclusion

This systematic review and meta-analysis reveals high prevalence of post-TB persistent symptoms, spirometric functional impairment and radiologic structural abnormalities in individuals living in LMICs. Burden of PTLD is highest in SE Asia followed by Africa, symptomatic PTLD is higher in rural compared to urban settings, and HIV infected persons have lower prevalence of PTLD compared to HIV uninfected persons. There is paucity of data on TB sequelae in children, and on the effect of biomass smoke exposure on post-TB lung health in LMICs.

Children and adolescents should be prioritised for inclusion in future research especially those below age 14 years, to provide insight into the magnitude and risk factors for post-TB sequelae in this highly vulnerable population. There is need to develop structured approaches within our health services for identification and management of this vulnerable population of TB survivors with residual lung disease after completion of TB treatment.

## Supporting information

**S1 Fig. Comparison of prevalence of persistent respiratory symptoms to prevalence of abnormal spirometry.**
(TIF)

**S2 Fig.** a. Prevalence of persistent respiratory symptoms across WHO regions. b. Prevalence of persistent respiratory symptoms across income levels. c. Prevalence of persistent respiratory symptoms across age groups.
(TIF)

**S3 Fig.** a. Prevalence of abnormal spirometry across WHO regions. b. Prevalence of abnormal spirometry across income levels. c. Prevalence of abnormal spirometry in rural, mixed, and urban settings. d. Prevalence of abnormal spirometry across age groups.
(TIF)

**S4 Fig.** a. Prevalence of PTLD as a composite outcome in female and male subgroups. b. Prevalence of PTLD as a composite outcome across age groups. c. Prevalence of PTLD as a composite outcome across income levels. d. Prevalence of PTLD as a composite outcome in urban, rural, and mixed settings.
(TIF)

**S1 Table. Search terms used for systematic review on post-tuberculosis lung disease in low- and middle-income countries.**
(PDF)

**S1 Data. Dataset PTLD.**
(XLSX)

## Acknowledgments

We would like to thank Dr Wanini Edemba and Peter M. Obimbo for their contribution to this work. Peter M. Obimbo assisted with data organisation and analysis, and formatting of tables for the manuscript while Dr Wanini Edemba participated in abstraction and cleaning of some of the clinical data.

## Author Contributions

**Conceptualization:** Elizabeth Maleche-Obimbo, Walter Jaoko, Fredrick Were, Stephen M. Graham.

**Data curation:** Lynette Njeri, Moses Mburu.

**Formal analysis:** Elizabeth Maleche-Obimbo, Mercy Atieno Odhiambo, Moses Mburu.

**Investigation:** Elizabeth Maleche-Obimbo, Mercy Atieno Odhiambo, Lynette Njeri, Moses Mburu.

**Methodology:** Elizabeth Maleche-Obimbo, Mercy Atieno Odhiambo, Moses Mburu.

**Project administration:** Elizabeth Maleche-Obimbo, Mercy Atieno Odhiambo.

**Supervision:** Fredrick Were, Stephen M. Graham.

**Writing – original draft:** Elizabeth Maleche-Obimbo, Mercy Atieno Odhiambo, Lynette Njeri.

**Writing – review & editing:** Walter Jaoko, Fredrick Were, Stephen M. Graham.

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
