## [Decision Letter · Decision Letter 0]

14 Jul 2022

PGPH-D-22-01006

Magnitude and factors associated with post-tuberculosis lung disease in low- and middle-income settings: a systematic review and meta-analysis

Dear Dr. Maleche-Obimbo,

Thank you for submitting your manuscript to PLOS Global Public Health. After careful consideration, we feel that it has merit but does not fully meet PLOS Global Public Health’s publication criteria as it currently stands. Therefore, we invite you to submit a revised version of the manuscript that addresses the points raised during the review process.

We look forward to receiving your revised manuscript.

Kind regards,

Julia Robinson

Staff Editor

Journal Requirements:

1. Please amend your online Financial Disclosure statement. If you did not receive any funding for this study, please simply state: “The authors received no specific funding for this work.”

2. Please update your online Competing Interests statement. If you have no competing interests to declare, please state: “The authors have declared that no competing interests exist.”

3. In the online submission form, you indicated that your data will be submitted to a repository upon acceptance. We strongly recommend all authors deposit their data before acceptance, as the process can be lengthy and hold up publication timelines. Please note that, though access restrictions are acceptable now, your entire data will need to be made freely accessible if your manuscript is accepted for publication. This policy applies to all data except where public deposition would breach compliance with the protocol approved by your research ethics board. If you are unable to adhere to our open data policy, please kindly revise your statement to explain your reasoning and we will seek the editor's input on an exemption. Please be assured that, once you have provided your new statement, the assessment of your exemption will not hold up the peer review process.

4. We ask that a manuscript source file is provided at Revision. Please upload your manuscript file as a .doc, .docx, or .rtf.

5. Please update your Supporting Information list of legends to include all supplementary files.

Additional Editor Comments (if provided):

Reviewers' comments:

Reviewer's Responses to Questions

**Comments to the Author**

1. Does this manuscript meet PLOS Global Public Health’s publication criteria? Is the manuscript technically sound, and do the data support the conclusions? The manuscript must describe methodologically and ethically rigorous research with conclusions that are appropriately drawn based on the data presented.

Reviewer #1: Yes

Reviewer #2: Yes

2. Has the statistical analysis been performed appropriately and rigorously?

Reviewer #1: Yes

Reviewer #2: Yes

3. Have the authors made all data underlying the findings in their manuscript fully available (please refer to the Data Availability Statement at the start of the manuscript PDF file)?

Reviewer #1: No

Reviewer #2: Yes

4. Is the manuscript presented in an intelligible fashion and written in standard English?

Reviewer #1: Yes

Reviewer #2: Yes

5. Review Comments to the Author

Reviewer #1: Title: Magnitude and factors associated with post-tuberculosis lung disease in low and middle-income settings: a systematic review and meta-analysis

• Is it settings or countries?

Abstract:

• The objective is merged with the methods in the abstract. It could be better to move it to the background with the justifications of conducting this review.

• Is it lower or low income countries? Please use the appropriate word consistently throughout the manuscript

• Statistical methods and the software used to compute the magnitude are not described. Try to describe some of the statistical methods and the selected effect sizes in methods part.

• Details of characteristics of included studies were included in the results. Focus on the main findings addressing the objectives.

• Keywords are too much. Please reduce them to the recommended

Introduction:

• Paragraph 1: TB can be also caused by other species of Mycobacterium and if you didn’t use terms that can exclude them, it will be confusing. So it may be better to use “usually caused by MTB” or “mostly caused by MTB”.

Methods:

• Literature search: the search terms used are narrow and missed important key term tuberculosis.

• Eligibility criteria: you stated that only articles written in English, or those with an English translation were eligible for inclusion; but you have described above in the literature search that no language restrictions was not imposed. How this contrasting information should be contextualized? If there are studies published in languages other than English, would you require translations or will you (the reviewers) translate them?

• Study selection: what quality of studies will be included? What is the importance of evaluating the quality of studies using Newcastle-Ottawa scale if poor quality studies are also included in your review? Is any quality of study with outcome of interest be selected? Is any sample size of studies be included? How many study participants are required from a single study to be included in your review? Brief description of these variables are required.

• Statistical analysis: which effect sizes are used to identify factors associated with PTLD?

Results:

• Eventually identified 102 eligible studies and subjected these to full paper review for eligibility. What does it mean? Have you identified 102 studies for eligibility or inclusion? It needs revision.

• 18 studies were excluded from your study because they are from high income countries. How it comes to full article review as your eligibility criteria stated that patients must have been living and studies conducted from low or middle income countries?

• Description of studies: para-2: The sample size was expressed in “subjects” but it is commonly advised to use “participants” or “patients” in human studies.

• Table 1b: what is the importance of expressing gender in terms of female (%)?

Discussion:

• The review and meta-analysis quantitatively reveal a high burden of post-TB lung disease among persons living in lower and lower-middle income settings. This lacks consistency of using LMIC. Are they “lower and lower middle” or “low and middle income?” Use these terms consistently.

Reviewer #2: Obimbo et al provide an excellent systematic review and metanalysis of studies that provided data on the magnitude ( incidence /prevalence ) of post TB lung disease in low- and middle-income countries. Data obtained from 32 studies and involving 6,050 patients revealed high rates of lung function impairment ( pooled prevalence 46.7%, 95% CI 36.9-56.7%, from 2,701 patients who had this information), persistent symptoms (pooled prevalence 41.0,% 95% CI 29.39-53.70% , from 3, 811 patients) and radiological abnormalities (pooled prevalence 64.63%, 95% CI 39.54 -83.63%, from 2,857 patients). The presence of any one of these measures of PTLD was found in 42.7% of the 6,050 patients ( 95% CI 34.64-51.14.)

There were regional differences in the magnitude of PTLD with the WHO regions of Western Pacific and Southeast Asia having the highest rates. Risk factor analysis revealed living or receiving treatment from a rural setting, being HIV seronegative and being older to be associated with higher rates of PTLD. There were only two studies that provided information on PTLD following treatment of multidrug resistant TB, both of which showed high rates of post TB sequelae more than 90% of patients.

The paper is well written. There are only minor issues that the authors may want to address:

1. In the introduction section it is stated that there were 10 million people who were diagnosed to have TB in 2019 of whom 1.2 million died of the disease. This is an erroneous interpretation of the information provided by the WHO global TB report. The figures provided in the WHO global TB report are estimates of the global incidence of TB (10 million people in 2019) and global mortality (1.2 million people) from the disease. In 2019, only about 7 million people were diagnosed to have TB and were reported to national TB programs and consequently reported to WHO leaving a TB notification gap of about 3 million people. The deaths from TB in that year are computed from both the notified and unnotified groups.

2. It may be good to present the data on incidence and prevalence separately. These two epidemiologic parameters provide significantly different information with different implications for programming for post TB care.

3. I am wondering why statistical measures of association (odds ratios or relative risk) are not provided for the risk factor analysis for PTLD.

4. I note that there are a few sentences on page 29 that are have grammatical errors

6. PLOS authors have the option to publish the peer review history of their article (what does this mean?). If published, this will include your full peer review and any attached files.

**Do you want your identity to be public for this peer review?** For information about this choice, including consent withdrawal, please see our Privacy Policy.

Reviewer #1: **Yes: **Addisu Melese

Reviewer #2: **Yes: **Jeremiah Chakaya

---

## [Decision Letter · Decision Letter 1]

30 Aug 2022

PGPH-D-22-01006R1

Magnitude and factors associated with post-tuberculosis lung disease in low and middle income countries: a systematic review and meta-analysis

Dear Dr. Maleche Obimbo,

Thank you for submitting your manuscript to PLOS Global Public Health. After careful consideration, we feel that it has merit but does not fully meet PLOS Global Public Health’s publication criteria as it currently stands. Therefore, we invite you to submit a revised version of the manuscript that addresses the points raised during the review process.

We look forward to receiving your revised manuscript.

Kind regards,

Lucy Chimoyi

Academic Editor

Journal Requirements:

1. Please ensure that the Title in your manuscript file and the Title provided in your online submission form are the same.

Additional Editor Comments (if provided):

Thank you for taking time to address the reviewers responses. This is a very well written piece of work.

There is one more comment from the reviewers that requires your attention.

Reviewers' comments:

Reviewer's Responses to Questions

**Comments to the Author**

1. If the authors have adequately addressed your comments raised in a previous round of review and you feel that this manuscript is now acceptable for publication, you may indicate that here to bypass the “Comments to the Author” section, enter your conflict of interest statement in the “Confidential to Editor” section, and submit your "Accept" recommendation.

Reviewer #1: All comments have been addressed

2. Does this manuscript meet PLOS Global Public Health’s publication criteria? Is the manuscript technically sound, and do the data support the conclusions? The manuscript must describe methodologically and ethically rigorous research with conclusions that are appropriately drawn based on the data presented.

Reviewer #1: Yes

3. Has the statistical analysis been performed appropriately and rigorously?

Reviewer #1: Yes

4. Have the authors made all data underlying the findings in their manuscript fully available (please refer to the Data Availability Statement at the start of the manuscript PDF file)?

Reviewer #1: Yes

5. Is the manuscript presented in an intelligible fashion and written in standard English?

Reviewer #1: Yes

6. Review Comments to the Author

Reviewer #1: Thank you for your response to almost all my comments.

But the authors response to my one comment on the cause of TB is not yet well addressed. You said that TB is caused by only MTB. But the literature and other relevant credible sources described that TB is caused by M. tuberculosis complex that comprised of MTB, M. bovis, M. africanum and other related species.The mycobacterial species that occurs in human and belong to M. tuberculosis complex include M. tuberculosis, M. bovis, M. bovis BCG and M. africanum: all species are capable of causing tuberculosis. You can check at https://www.nationaljewish.org/conditions/tuberculosis-tb/causes. As the authors didn't specifically describe the disease caused is a type of pulmonary TB, it will be difficult to agree with and it is misleading to declare TB is only caused by MTB. Or the authors should describe it as "post-pulmonary tuberculosis" in the title or elsewhere in the mansucript.

7. PLOS authors have the option to publish the peer review history of their article (what does this mean?). If published, this will include your full peer review and any attached files.

**Do you want your identity to be public for this peer review?** For information about this choice, including consent withdrawal, please see our Privacy Policy.

Reviewer #1: **Yes: **Addisu Melese

---

## [Decision Letter · Decision Letter 2]

2 Oct 2022

PGPH-D-22-01006R2

Magnitude and factors associated with post-tuberculosis lung disease in low- and middle-income countries: a systematic review and meta-analysis

Dear Dr. Maleche-Obimbo,

Thank you for submitting your manuscript to PLOS Global Public Health. After careful consideration, we feel that it has merit but still need some minor reviews as it does not fully meet PLOS Global Public Health’s publication criteria as it currently stands. Therefore, we invite you to submit a revised version of the manuscript that addresses the points raised during the review process.

We look forward to receiving your revised manuscript.

Kind regards,

Anete Trajman

Academic Editor

Journal Requirements:

Additional Editor Comments (if provided):

Thank you for revising your manuscript. Besides the reviewer suggestion, please include in your discussion an explanation of why you decided to pool studies despite the high heterogeneity.

In your next revised version, please ensure that you have edited and minutiously reviewed the entire text, as some minor editorial issues remain (final dot missing at the end of sentences, incomplete sentence page 26 and others).

Reviewers' comments:

Reviewer's Responses to Questions

**Comments to the Author**

1. If the authors have adequately addressed your comments raised in a previous round of review and you feel that this manuscript is now acceptable for publication, you may indicate that here to bypass the “Comments to the Author” section, enter your conflict of interest statement in the “Confidential to Editor” section, and submit your "Accept" recommendation.

Reviewer #1: All comments have been addressed

Reviewer #3: All comments have been addressed

2. Does this manuscript meet PLOS Global Public Health’s publication criteria? Is the manuscript technically sound, and do the data support the conclusions? The manuscript must describe methodologically and ethically rigorous research with conclusions that are appropriately drawn based on the data presented.

Reviewer #1: Yes

Reviewer #3: Yes

3. Has the statistical analysis been performed appropriately and rigorously?

Reviewer #1: Yes

Reviewer #3: Yes

4. Have the authors made all data underlying the findings in their manuscript fully available (please refer to the Data Availability Statement at the start of the manuscript PDF file)?

Reviewer #1: Yes

Reviewer #3: Yes

5. Is the manuscript presented in an intelligible fashion and written in standard English?

Reviewer #1: Yes

Reviewer #3: Yes

6. Review Comments to the Author

Reviewer #1: (No Response)

Reviewer #3: After the first revisions, the manuscript improved significantly. However, there is an important recent document that should be cited and discussed by the authors:

Migliori GB et al. Clinical standards for the assessment, management and rehabilitation of post-TB lung disease. IJTLD 2021; 25(10):797-813. doi: 10.5588/ijtld.21.0425.

7. PLOS authors have the option to publish the peer review history of their article (what does this mean?). If published, this will include your full peer review and any attached files.

**Do you want your identity to be public for this peer review?** For information about this choice, including consent withdrawal, please see our Privacy Policy.

Reviewer #1: **Yes: **Addisu Melese

Reviewer #3: No

---

## [Editor Report · Decision Letter 3]

7 Nov 2022

Magnitude and factors associated with post-tuberculosis lung disease in low- and middle-income countries: a systematic review and meta-analysis

PGPH-D-22-01006R3

Dear Prof Maleche-Obimbo,

We are pleased to inform you that your manuscript 'Magnitude and factors associated with post-tuberculosis lung disease in low- and middle-income countries: a systematic review and meta-analysis' has been provisionally accepted for publication in PLOS Global Public Health.

Best regards,

Anete Trajman

Academic Editor